# GPT Deciphering Fedspeak: Quantifying Dissent Among Hawks and Doves

**Denis Peskoff**
Office of Population Research
Princeton University
dp2896@princeton.edu

**Adam Visokay**
Sociology
University of Washington
avisokay@uw.edu

**Sander Schulhoff**
Computer Science
University of Maryland
sschulho@umd.edu

**Benjamin Wachspress**
Computer Science
Princeton University
bjw6@princeton.edu

**Alan Blinder**
Economics
Princeton University
blinder@princeton.edu

**Brandon M. Stewart**
Sociology and OPR
Princeton University
bms4@princeton.edu

## Abstract

Markets and policymakers around the world hang on the consequential monetary policy decisions made by the Federal Open Market Committee (FOMC). Publicly available textual documentation of their meetings provides insight into members' attitudes about the economy. We use GPT-4 to quantify dissent among members on the topic of inflation. We find that transcripts and minutes reflect the diversity of member views about the macroeconomic outlook in a way that is lost or omitted from the public statements. In fact, diverging opinions that shed light upon the committee's "true" attitudes are almost entirely omitted from the final statements. Hence, we argue that forecasting FOMC sentiment based solely on statements will not sufficiently reflect dissent among the hawks and doves.

## 1 The Road to FOMC Transparency

The Federal Open Market Committee (FOMC) is responsible for controlling inflation in the United States, using instruments which dramatically affect the housing and financial markets, among others. For most of the 20[th] century, conventional wisdom held that monetary policy is most effective when decision-making was shrouded in secrecy; the tight-lipped Alan Greenspan, a past chairman of the Fed, quipped about *"learning to mumble with great incoherence."* But times change.

Blinder et al. (2008) show how the emergence of greater transparency and strategic communication became an important feature of 21[st] century central banking. Fed communication is now an integral component of monetary policy, and "Fed watchers" dote on every word. The FOMC first started releasing public statements following their meetings in February 1994. This meager documentation grew and now consists of three types for each official meeting: carefully produced and highly stylized

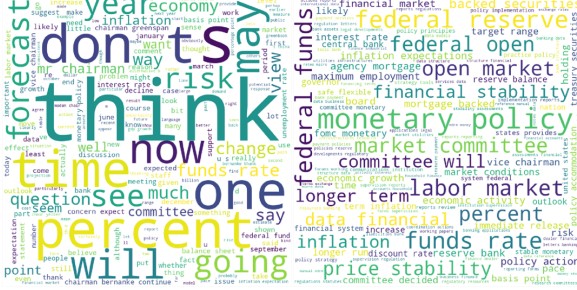

Figure 1: The transcripts (left) contains opinions and disagreements but statements (right) are concise. We analyze both datasets using GPT-4 prompting.

one page statements are released immediately after each FOMC meeting, followed about three weeks later by lengthier minutes, and finally five years later by full, verbatim transcripts. Subsequently this triplet is referred to as documents. We find minutes closely reflect the content of transcripts, so to avoid redundancy, we focus our analysis on transcripts and statements.

Increased FOMC communication has prompted social science research spanning the disciplines of economics, sociology, finance and political science (Section 2.1). Financial market participants are also keenly interested. Billions, if not trillions of dollars are traded on the Fed's words. The interpretations—right or wrong—of what the FOMC "really means" move markets and affect the economy. However, relying upon documents as data in the social sciences is a challenge due to the lack of structure and the cost of annotation (Grimmer and Stewart, 2013; Gentzkow et al., 2019; Ash and Hansen, 2023).

Hansen and Kazinnik (2023) show that Generative Pre-training Transformer (GPT) models outperform a suite of commonly used NLP methods on text quantification. Motivated by these results, we set out to quantify the language of the FOMC using GPT-4 (OpenAI, 2023) by preparing a combined data set of FOMC documents from 1994-

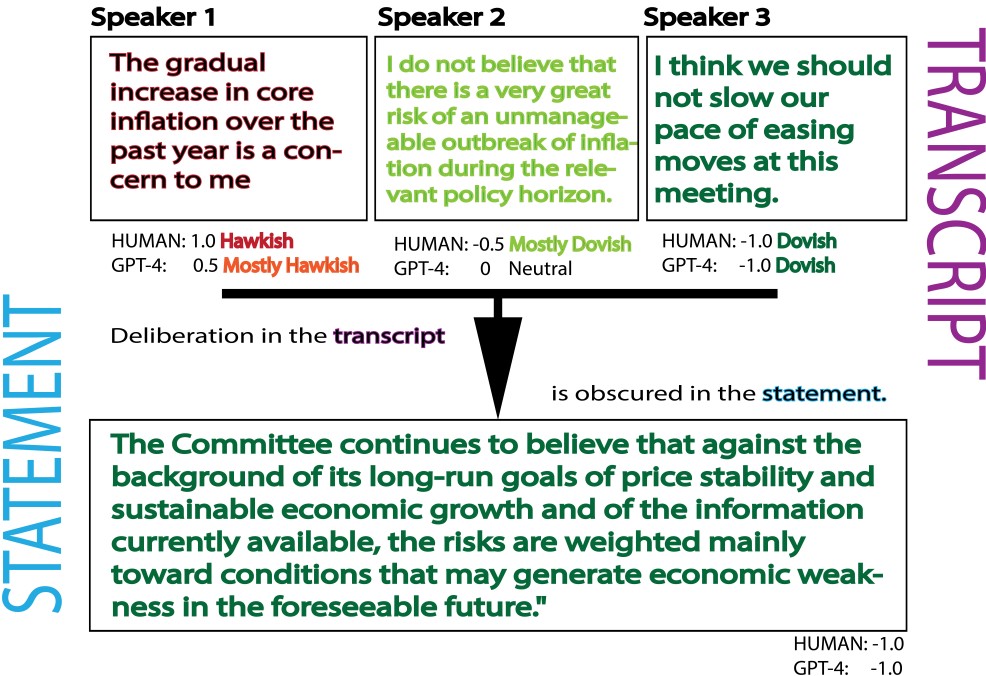

Figure 2: Text taken from an FOMC meeting on December 11, 2001. A dovish statement does not reflect the hawkish sentiment of Speaker 1. GPT-4 can quantify dissent lost from transcripts to statements.

2016 (Section 2).[1] We conclude that transcripts contain more dissent than statements (Section 3).

## 2 FOMC Data: Transcript to Statement

The FOMC normally meets eight times per year in order to assess current economic conditions, ultimately deciding upon the path for monetary policy. We aggregate and release the official publicly available text documenting these deliberations by the Fed as an aligned corpora of documents from 1994 to 2016.[2] These text documents are similar in content, but transcripts and statements are dramatically different in style and detail (Figure 1).[3] For our purposes, the statements required no pre-processing. For the transcripts, we use regular expressions to partition and then re-aggregate the text by each unique speaker. See Appendix D for an example of each document type.

### 2.1 Lessons from Social Science

Past work has used at most one form of FOMC meeting documentation, but *rarely multiple in conjunction*. For example, in the finance literature,

[1] https://github.com/DenisPeskoff/FedNLP
[2] https://www.federalreserve.gov/monetarypolicy/fomc_historical.htm
[3] Transcripts only became being released in 1994 and due to the 5 year lag the latest transcripts are not yet available.

Mazis and Tsekrekos (2017) apply Latent Semantic Analysis to FOMC statements to identify the main "themes" used by the committee and how well they explain variation in treasury yields. Gu et al. (2022) use minutes to investigate how the tonality of committee deliberations impacts subsequent stock market valuations. Political scientists use transcripts to estimate committee members' preferences on inflation and unemployment (Baerg and Lowe, 2020). Economists have assessed the role of communication in achieving monetary policy objectives by looking at similar documents (Romer and Romer, 2004; Handlan, 2020). Hüpper and Kempa (2023) investigate the extent to which shifting inflation focus is reflected in full transcripts. Edison and Carcel (2021) apply Latent Dirichlet Allocation (LDA) to transcripts to detect the evolution of prominent topics. Hansen et al. (2017) use LDA to quantify transcripts and identify how transparency affects the committee's deliberations.

### 2.2 Hawks and Doves

We take the transcripts to best represent FOMC members' underlying attitudes and think of the statements as stylized representations of what they wish to communicate publicly. To identify disagreement, we need to go beyond the statements and look closely at the language employed by members

in their remarks throughout meeting transcripts. Dissenting votes are rare because of a strong historical norm: members dissent only if they feel very strongly that the committee's decision is wrong. Modest disagreements do not merit dissent.[4] That said, members of the committee do frequently voice detectable disagreements with one another at meetings. Disagreements frequently concern the state of the economy, the outlook for inflation, and many other things, including where the range for the federal funds rate should be set that day. Such debate and deliberation among members is a routine and productive element of the meetings. These disagreements are more clearly expressed in the transcripts. Daniel Tarullo's comments in a 2016 transcript illustrate the point:

> "it is institutionally important for us to project an ability to agree, even if only at a fairly high level, and that is why I abstained rather than dissented over each of the past several years [...] I have gone out of my way in the past four years not to highlight publicly my points of difference with the statement."

### 2.3 Manual Analysis to Create a Gold Label

Dissent amongst speakers is normally concentrated on the discussion of the economic and financial situation of the U.S, specifically inflation targeting. For example, in the January 2016 meeting transcript, the committee discusses their 2 percent inflation projection in the context of factors such as oil prices, the job market, and the Chinese economy. Many of the members argue that the inflation projection of 2 percent will not be accurate, while others who do support the 2 percent projection qualify their support with varying degrees of uncertainty. In this meeting, President Mester concludes,

> "My reasonable confidence that inflation will gradually return to our objective over time recognizes there is and has always been large uncertainty regarding inflation forecasts."

While Mr. Tarullo argues in opposition,

> "... I didn't have reasonable confidence that inflation would rise to 2 percent.

---

[4]Notably, this is not true of the monetary policy committees of other nations; on some of them, dissent is common—even expected (Tillmann, 2021).

> Nothing since then has increased my confidence. To the contrary, a few more doubts have crept in."

If the meeting statement was an accurate representation of what transpired at the meeting, it would follow that the uncertainty of the committee regarding their inflation forecast would be communicated. Instead, the diverging individual opinions are omitted in the final statement, where:

> "The Committee currently expects that, with gradual adjustments in the stance of monetary policy, economic activity will expand at a moderate pace and labor market indicators will continue to strengthen. Inflation is expected to remain low in the near term, in part because of the further declines in energy prices, but to rise to 2 percent over the medium term as the transitory effects of declines in energy and import prices dissipate and the labor market strengthens"

We apply this level of granular analysis to create a gold label for statements, classifying each one according to Hawk and Dove definitions proposed by Hansen and Kazinnik (2023): Dovish (-1.0), Mostly Dovish (-0.5), Neutral (0), Mostly Hawkish (.5), Hawkish (1.0). A trained undergraduate does a first pass and escalates any borderline cases to a FOMC expert for adjudication. We manually review the January 2016 transcript and pair it with a simple computational analysis, which finds that most dissent at that time surrounded topics relating to inflation (Appendix C).

### 2.4 GPT-4 "Reads" Terse Documents

GPT-4, and Large Language Models more broadly, are a suitable tool for rapid linguistic processing at scale. We produce three different measurements of hawk/dove sentiment using statements and two using transcripts. Hansen and Kazinnik (2023) use GPT-3 to quantify 500 sentences selected uniformly at random from FOMC statements between 2010 and 2020. We extend this by using GPT-4 to quantify **all** 3728 sentences in statements from 1994 to 2016 (Appendix B). A limitation of this approach is that the holistic sentiment of the meeting is not captured because each sentence is scored independently—without context.

The first statement measurement we propose is to simply take an unweighted mean of all individual

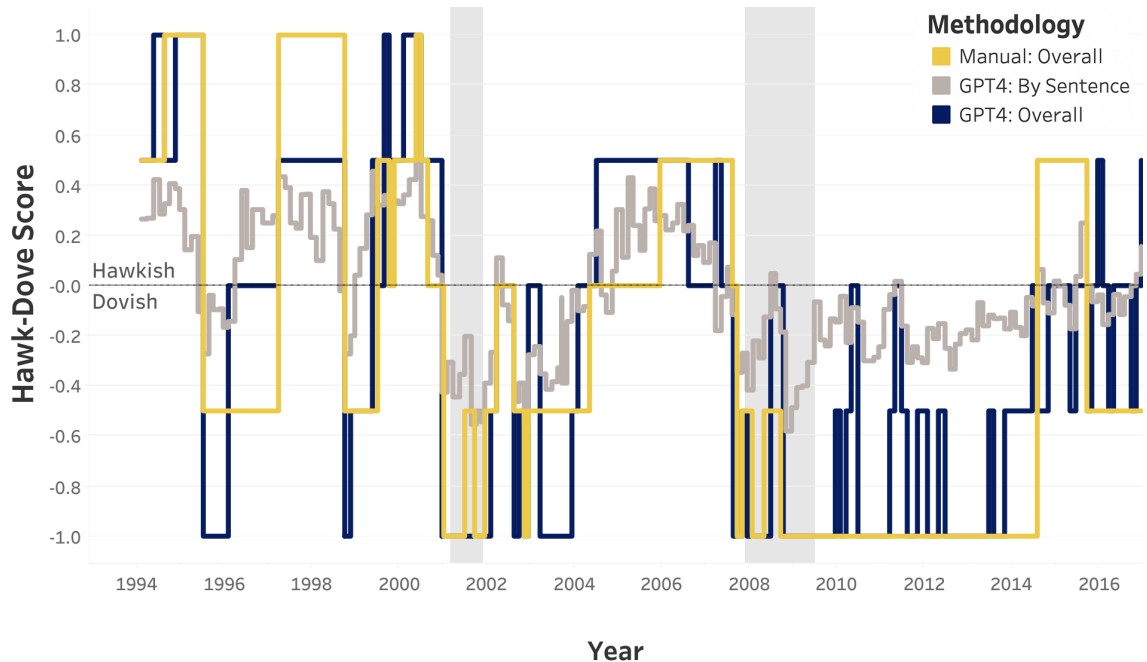

Figure 3: Our analysis of statements finds that averaging at the sentence-level (gray) loses information since the average sentence is Neutral. Ingesting the overall statement (blue) better mirrors the manual gold label (gold).

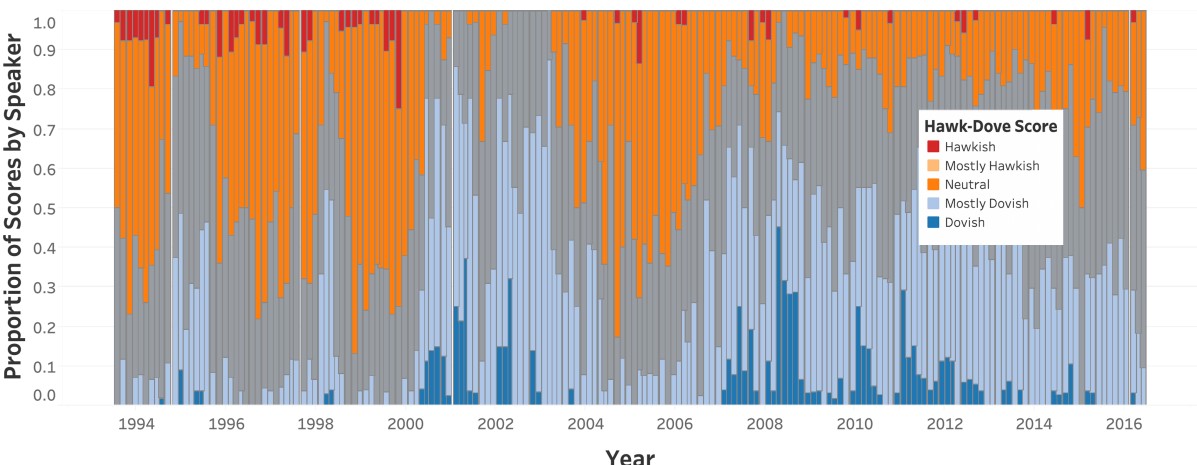

Figure 4: Our analysis of transcripts with GPT-4 at the speaker level shows that meetings consistently have dissenting opinions. Similarly to the previous figure, 2001 to 2004 is dovish (blue), while 2004 to 2006 is hawkish (orange).

sentence scores for each meeting. Because most sentences have nothing to do with inflation (62% of sentences scored as Neutral), we hypothesize that this method has further limitations. We resolve this by ingesting each *entire* statement into our GPT-4 prompt. Both of these measurements, along with manual gold labels, can be seen in Figure 3.

For the final statement measurement, we construct a logit-scaled score (Lowe et al., 2011),

$$\theta^{(L)} = \log\left(\frac{Hawk + 0.5}{Dove + 0.5}\right)$$

where *Hawk* and *Dove* are the sums of the hawk-ish and dovish scores, respectively. In this approach $\theta^{(L)}$ ignores sentences scored as Neutral, placing more emphasis on the *relative* rather than *absolute* differences between hawkish and dovish sentiment. Furthermore, since $\theta^{(L)}$ has no predefined end points, this allows us to generate positions at any level of extremity, which more appropriately reflects the outlier meetings.

When measuring hawk/dove sentiment using transcripts, the vast amount of text adds an additional challenge. Rather than evaluating a 100+ page transcript at the sentence level, we instead evaluate transcripts at the speaker level. That is to

say, we use GPT-4 to quantify hawk/dove sentiment for each distinct speaker within each transcript, and then aggregate those by meeting date. In contrast to the 3728 individual sentence observations across all statements, we evaluate 5691 speaker observations across all transcripts. We then follow the same steps outlined above to calculate unweighted average and logit-scaled scores.

The logit-scaled measurements for transcripts and statements track one another quite closely over most of the years, demonstrating that GPT-4 is effective at identifying similar content across document types of dramatically different length and style. It is worth noting, however, that the logit-scaled transcript scores have larger extremes than statements, especially the upper bound: (-3.66, 3.37) and (-3.40, 2.40), respectively. This underscores how the FOMC devotes considerable attention to curating their communication strategy to convey confidence and unity despite dissent among the hawks and doves. The logit-scaled scores begin to diverge in 2012, with transcripts trending more neutral/hawkish while statements remain mostly dovish. Hence, we propose a direct comparison.

# 3 LLMs Quantify Economic Text

## 3.1 Measuring Dissent

We can use the sentence-level statement and speaker-level transcript scores from GPT-4 to compute a measure of dissent for each meeting using the following algorithm:

1. From the list of scores for each meeting, count the number of hawkish/mostly hawkish and the number of dovish/mostly dovish scores.
2. If there is at least one hawkish score and at least one dovish score within the same meeting, assign Dissent = 1. Else, Dissent = 0.

We find that **47%** of statements and **82%** of transcripts contain dissent. We also compute the conditional probability of a transcript containing dissent given the associated statement binary, $P(T = 1|S = 1)$ and $P(T = 1|S = 0)$. We find that when a statement contains dissent, $P(T = 1|S = 1)$, the transcript agrees more than **97%** of the time. However, for statements scored as having no dissent, $P(T = 1|S = 0)$ we find that more than **69%** of associated transcripts are scored as containing dissent. This means that for the 53% of statements that don't show signs of dissenting opinions, there is likely dissent in the transcript as evidenced by the speaker-level hawk/dove scores.

## 3.2 Conclusion and Next Steps

Our method of ingesting the entire statement for an aggregate prediction better captures the extremes, which more closely mirror the gold label human annotation and suggests that Large Language Models can avoid the noise in this nuanced context. The $F_1$ score for this comparison is 0.57. While this is rather low as a measure of model "fit", it is important to note that the results rarely flip sentiment (from hawkish to dovish, or mostly hawkish to mostly dovish), rather, it just seems to mostly disagree on adjacent categories. See Figure 3 for a visual comparison of the sentence-level, entire text, and manual scores. Of note, the inconsistent provision of statements and relatively high volatility in hawk/dove sentiment before 2000 is consistent with Meade and Stasavage (2008) and Hansen et al. (2017) who have also studied the 1993 change in FOMC communication strategy. We demonstrate that GPT-4 can identify the extremes in dissenting hawk and dove perspectives despite the indications of a clear consensus in the statements. This empirical finding supports our manual analysis.

While we focus on transcripts and statements, future work may consider an even more fine grained analysis, incorporating minutes as well. We found the content of the minutes to more closely resemble the transcripts than the statements, but differences do exist and remain underexplored.

Additionally, we note that GPT-4 scores made more neutral predictions than the gold standard manual labels. To improve upon this, we created a balanced few-shot example using sentences from FOMC statements not included in our sample — meetings since 2020. This marginally improved the prediction "fit" ($F_1$ of 57% to 58%), but we expect that this could be improved much further with additional prompt engineering.

GPT-4 is able to quickly quantify stylized economic text. Our results from quantifying dissent support the hypothesis that dissenting opinions on the topic of inflation omitted from FOMC statements can be found in the associated transcripts. As LLMs continue to improve, we expect that it will be possible to study even more nuanced questions than the ones we answer here.

## Limitations

Substantively, strategic signaling in the FOMC is a challenging topic and this is only an initial investigation. Dissent does not have clear ground truth

labels and thus we are reliant on human judgment and our team's substantive expertise on monetary policy. Finally, as with much current research, our work relies on OpenAI's GPT API, which poses challenges to computational reproducibility, as it relies on the stability of an external system that we cannot control.

## Ethics Statement

The FOMC is a high-stakes body whose activities are already subject to substantial scrutiny. There is some ethical risk in exploring linguistic signals of hidden information. For example, based on the substantive literature we believe that dissent is intentionally signalled in meetings in order to set up future discussions or lay claims to particular positions. Nevertheless, attribution of intention (as implied by 'dissent') always involves some level of error that could be uncomfortable to meeting participants who feel mischaracterized. We also emphasize that our approach to capturing dissent would not be appropriate to use outside this specific context without careful validation. Finally, by making the FOMC data more easily available to the NLP community, we also assume some ethical responsibility for the potential uses of that data (see e.g. Peng et al., 2021). We spend under $1500 on computation and under $1000 on annotation and believe our results to be reasonably reproducible. We feel that these concerns are ultimately minor given that all participants are public officials who knew their transcripts would ultimately be released.

## Acknowledgements

This material is based upon work supported by the National Science Foundation under Grant #2127309 to the Computing Research Association for the CIFellows 2021 Project. We thank the Initiative for Data-Driven Social Science at Princeton University for grant support and OpenAI for technical support. Visokay is supported by the National Institute of Mental Health of the NIH under Award Number #DP2MH122405 and by the Center for Statistics and the Social Sciences at the University of Washington.

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

# A  Appendix

Our appendix contains the GPT prompt we used, a section on our computational analysis, and an example of the statement (Figure 6), transcript (Figure 7), and minutes (Figure 8) for the January 29-30, 2008 meetings.

# B  GPT-4

We used GPT-4 heavily for our experiments and analysis. Recent work (OpenAI, 2023; Liu et al., 2023; Guan et al., 2023) that successfully uses GPT-4 for classification gave us confidence in its quality. Additionally, a human expert on our team examined GPT-4 generated labels, and found that in a sample of 25, our expert agreed with 19 labels with high confidence, and 22 labels with at least moderate confidence.

## B.1  Methodology

We used variants of a single prompt template for all of our tasks. It contains the relevant labels (Hawkish, Dovish, etc.) as well as their definitions. It includes space for some INPUT and asks which label best applies to the input. When processing different documents, such as transcripts, we would switch out the word statements in the prompt for the appropriate document word.

For statements, we 0-shot prompted GPT-4 to label each statement as one of the five labels. We also ran an experiment in which we classified each sentence of each statements as one of the labels, then averaged the sentence scores to get to statement score. Finally, we reran this sentence classification with a 10-shot prompt. The prompt was similar to below, except with 10 examples of sentences and their classifications at the beginning.

For minutes, we 0-shot prompted GPT-4-32K to label each statement as one of the five labels.

For transcripts, we 0-shot prompted GPT-4-32K to examine all of each speakers speech, and provide each speaker a single label for each transcript.

For all API calls to OpenAI, we only modified the model (either GPT-4 or GPT-4-32K). We did not change any other settings.

```
prompt_template = """
<statement>
INPUT
</statement>
<labels>
```

```
Dovish : Strongly expresses a
    belief that the economy may be
growing too slowly and may need
    stimulus through mon−
etary policy.
Mostly dovish : Overall message
    expresses a belief that the
    economy may
be growing too slowly and may
    need stimulus through
monetary policy.
Neutral : Expresses neither a
    hawkish nor dovish view and is
mostly objective.
Mostly hawkish : Overall message
    expresses a belief that the
    economy is
growing too quickly and may need
    to be slowed down
through monetary policy.
Hawkish : Strongly expresses a
    belief that the economy is
    growing
too quickly and may need to be
    slowed down through monetary
    policy.
</labels>
Which label best applies applies
    to the statement (Dovish,
    Mostly Dovish , Neutral , Mostly
     Hawkish , Hawkish)?
"""
```

## C   Computational Analysis

We paired our manual review of the January 26-27, 2016 transcript with a computational analysis of dissent in the meeting. We stratified the meeting into nine topics, each corresponding to a portion of the transcript content. As a baseline, we counted the number of speakers in each section to see if this metric could reflect dissent. This technique, however, seemed to reflect the length of the conversation as opposed to the degree to which members disagreed with one another.

Our next approach was to do a sentiment analysis of each topic to see if the prevalence of negativity could indicate dissent. We supposed that negative sentiment would be high if the speakers opposed the stance of either other individuals or the committee as a whole. Using the VADER lexicon (Hutto and Gilbert, 2014), we calculated the

sentiment of each sentence within the nine topics. Since VADER is trained on web-based social media content, which is typically more abrupt than the formal language appearing in the FOMC transcript, we conducted the sentiment analysis by sentence to optimize the method's performance.

To analyze dissent more specifically, we computed the fraction of negative sentences in each topic. For this analysis, we set the threshold negativity score to be 0.1. That is, sentences with a negativity score of 0.1 or higher were classified as negative while all others were not. This number determined by manually reviewing what sentences were captured by varying thresholds and evaluating whether or not they conveyed dissent. When the threshold was set too low (0.05), four out of ten randomly selected sentences conveyed dissent. When set too high (0.15), seven out of ten randomly selected sentences conveyed dissent, but many sentences that indicated dissent were omitted. At the threshold of 0.1, still seven out of ten randomly selected sentences conveyed dissent, and more sentences that conveyed dissent were captured.

## D   Document Examples

See Figures 6, 7, and 8 for examples of the documents.

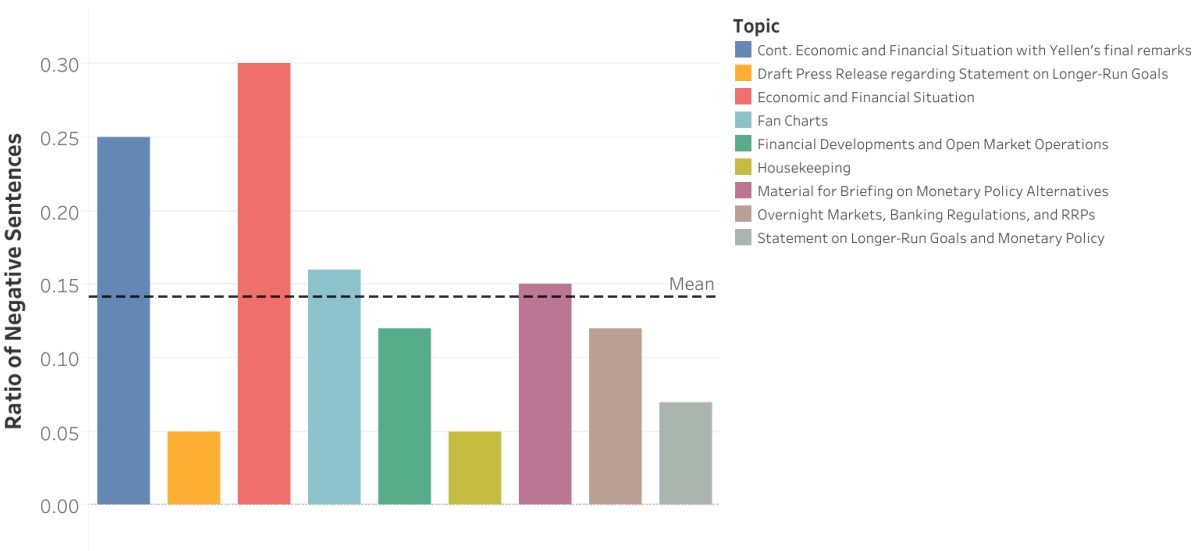

Figure 5: Discussion of inflation (Red and Blue topics) is more contentious than other topics and the average of all data (dotted line).

## Press Release

January 27, 2016

## Federal Reserve issues FOMC statement

For release at 2:00 p.m. EST

Share

Information received since the Federal Open Market Committee met in December suggests that labor market conditions improved further even as economic growth slowed late last year. Household spending and business fixed investment have been increasing at moderate rates in recent months, and the housing sector has improved further; however, net exports have been soft and inventory investment slowed. A range of recent labor market indicators, including strong job gains, points to some additional decline in underutilization of labor resources. Inflation has continued to run below the Committee's 2 percent longer-run objective, partly reflecting declines in energy prices and in prices of non-energy imports. Market-based measures of inflation compensation declined further; survey-based measures of longer-term inflation expectations are little changed, on balance, in recent months.

Consistent with its statutory mandate, the Committee seeks to foster maximum employment and price stability. The Committee currently expects that, with gradual adjustments in the stance of monetary policy, economic activity will expand at a moderate pace and labor market indicators will continue to strengthen. Inflation is expected to remain low in the near term, in part because of the further declines in energy prices, but to rise to 2 percent over the medium term as the transitory effects of declines in energy and import prices dissipate and the labor market strengthens further. The Committee is closely monitoring global economic and financial developments and is assessing their implications for the labor market and inflation,

Figure 6: Released statement for the January 26-27, 2016 meeting.

CHAIR YELLEN. So maybe it should be "the Committee."

MS. MESTER. Right.

CHAIR YELLEN. Let's try that again: "the Committee's inflation goal is symmetric, but believed that the statement should more clearly express that the Committee is primarily"—

MR. LACKER. Right.

MS. MESTER. Right.

MR. WILLIAMS. No.

MR. EVANS. You're starting to talk about Committee actions at that point.

PARTICIPANTS. It's the statement.

CHAIR YELLEN. "But believed that the statement should more clearly express that"— what?—"the Committee is primarily focused" or—

MR. LOCKHART. "That the Committee's primary focus be on expected future deviations."

MR. WILLIAMS. No.

MR. LACKER. Madam Chair, could we try the passive voice, because it avoids that. "It should more clearly express a focus on" or "a primary focus on."

MR. POWELL. Dan's language avoided this whole mess.

MR. WILLIAMS. Yes. We were good before.

MR. POWELL. "But believed the Committee's language is not adequately," or maybe "the statement language is not adequately focused on expected future deviations."

VICE CHAIRMAN DUDLEY. How about "sufficiently" rather than "adequately"?

CHAIR YELLEN. "That the statement language is not"—

MR. POWELL. —"sufficiently" or "adequately focused on expected"—et cetera.

Figure 7: A page of the January 26-27, 2016 meeting transcript that shows dissent.

Inflation, employment, and long-term interest rates fluctuate over time in response to economic and financial disturbances. Moreover, monetary policy actions tend to influence economic activity and prices with a lag. Therefore, the Committee's policy decisions reflect its longer-run goals, its medium-term outlook, and its assessments of the balance of risks, including risks to the financial system that could impede the attainment of the Committee's goals.

The inflation rate over the longer run is primarily determined by monetary policy, and hence the Committee has the ability to specify a longer-run goal for inflation. The Committee reaffirms its judgment that inflation at the rate of 2 percent, as measured by the annual change in the price index for personal consumption expenditures, is most consistent over the longer run with the Federal Reserve's statutory mandate. The Committee would be concerned if inflation were running persistently above or below this objective. Communicating this symmetric inflation goal clearly to the public helps keep longer-term inflation expectations firmly anchored, thereby fostering price stability and moderate long-term interest rates and enhancing the Committee's ability to promote maximum employment in the face of significant economic disturbances. The maximum level of employment is largely determined by nonmonetary factors that affect the structure and dynamics of the labor market. These factors may change over time and may not be directly measurable. Consequently, it would not be appropriate to specify a fixed goal for employment; rather, the Committee's policy decisions must be informed by assessments of the maximum level of employment, recognizing that such assessments are necessarily uncertain and subject to revision. The Committee considers a wide range of indicators in making these assessments. Information about Committee participants' estimates of the longer-run normal rates of output growth and unemployment is published four times per year in the FOMC's Summary of Economic Projections. For example, in the most recent projections, the median of FOMC participants' estimates of the longer-run normal rate of unemployment was 4.9 percent.

In setting monetary policy, the Committee seeks to mitigate deviations of inflation from its longer-run goal and deviations of employment from the Committee's assessments of its maximum level. These objectives are generally complementary. However, under circumstances in which the Committee judges that the objectives are not complementary, it follows a balanced approach in promoting them, taking into account the magnitude of the deviations and the potentially different time horizons over which employment and inflation are

projected to return to levels judged consistent with its mandate.

The Committee intends to reaffirm these principles and to make adjustments as appropriate at its annual organizational meeting each January."

All Committee members but one voted to adopt the revised statement. Although Mr. Bullard supported the statement without the changes and agreed that the Committee's inflation goal is symmetric, he dissented because he judged that the amended language was not sufficiently focused on expected future deviations of inflation from the 2 percent objective. In addition, because the Committee's past behavior had demonstrated the emphasis it places on expected future inflation, Mr. Bullard viewed the amended language as potentially confusing to the public.

**Developments in Financial Markets, Open Market Operations, and Policy Normalization**
The SOMA manager reported on developments in domestic and foreign financial markets, including changes in the expectations of market participants for the trajectory of monetary policy. The deputy manager followed with a briefing on money market developments and System open market operations conducted by the Open Market Desk during the period since the Committee met on December 15–16, 2015. The report included an assessment of the response of money market interest rates to the increase in the target range for the federal funds rate announced following the December meeting. Overall, the rate increase was implemented smoothly and money markets responded as anticipated. Take-up of overnight reverse repurchase agreement (ON RRP) operations over this period was consistent with that observed in the testing phase of operations over the second half of last year. The deputy manager also reviewed plans for reinvestment of the proceeds of upcoming maturations of SOMA holdings of Treasury securities, for small-value tests of various System operations and facilities during 2016, and for quarterly tests of the Term Deposit Facility.

The Committee then resumed its consideration of matters related to the System's reverse repurchase agreement (RRP) facilities, focusing in particular on the appropriate aggregate capacity of the ON RRP facility going forward. Previous communications had indicated that the Committee intended to allow aggregate capacity of the ON RRP facility to be temporarily elevated after policy firming had commenced to support monetary policy implementation and expected that it would be appropriate to reduce capacity fairly soon thereafter. A

Figure 8: A page of the January 26-27, 2016 meeting minutes that discusses inflation targeting.