# OpenReview forum: "GPT Deciphering Fedspeak: Quantifying Dissent Among Hawks and Doves"
_EMNLP/2023/Conference — EMNLP 2023 Findings_

### Official Review · Reviewer_gKYa · 2023-07-26

**Soundness:** 3

**Excitement:**

4: Strong: This paper deepens the understanding of some phenomenon or lowers the barriers to an existing research direction.

**Missing References:**

None

**Paper Topic And Main Contributions:**

The paper presents an application of LLMs to identifying dissent within FOMC transcripts, and use this method to explore the Hawkishness/Dovishness of FOMC policy attitudes over time

**Questions For The Authors:**

See questions embedded in the section above.

**Reasons To Accept:**

+ Important and novel question and data
+ Nice combination of domain and technical expertise

**Reasons To Reject:**

- Ultimately, I really don't understand how this was evaluated. Figure 3a) shows a visual result that I presume could be turned into some kind of quantitatve analysis with limited effort (although I'd think that one would want to account for time in that analysis, which makes things a bit trickier), but as it is I am just left to compare pairs of lines and trust intuition. I can see the authors point about extreme values, but I don't really understand why I couldn't just rescale the sentence-level measure and get the same effect
- Figure 3b) seems like maybe another evaluation, or just an analysis? If the former, I don't really know what is being evaluated. If the latter, more details on why differences between the transcript and statement measures emerges in specific years would be interesting (I get the gist from the rest of the paper about why we should see differences, but would be curious to hear more about specific years)

**Reproducibility:**

4: Could mostly reproduce the results, but there may be some variation because of sample variance or minor variations in their interpretation of the protocol or method.

**Reviewer Confidence:**

4: Quite sure. I tried to check the important points carefully. It's unlikely, though conceivable, that I missed something that should affect my ratings.

---

> ### Author Rebuttal · Authors · 2023-08-29
>
> We thank Reviewer 3 for their feedback.  We appreciate the acknowledgement of the novelty and for stressing the importance of pairing technical expertise with domain knowledge.
>
> Regarding Figure 3a, we appreciate the questions and concerns you raise. While the sentence-level method and the full document method are both measuring the same thing (hawk/dove sentiment), simply rescaling the former to try to capture the latter would not be appropriate. The contrast highlights an important result about which method most closely reflects gold standard human annotation. Figure 3a is an empirical assessment of our proposed methodology compared to Hansen and Kazinnik (2023)'s method which scored at the sentence level. Section 2.4 (*Lines 203-223*) currently explains this.  Furthermore, the fact that our preferred methodology considers every available meeting rather than a random sample of them, like the original work, is non-trivial (*Lines 194-199*).
>
> We agree with your suggestion to include a measure quantifying the deviation between the sentence-level measure à la Hansen and Kazinnik (2023) and our preferred measures that ingest the entire statement. We will include this measure in-line as part of our discussion of Figure 3a.
>
> Regarding Figure 3b, the equation on Line 214 is used to construct the logit-scaled hawk/dove score. This is neither a new evaluation nor analysis, simply a rescaling of the hawk/dove scores to better reflect extreme values. Re-scaling the scores in this way puts more emphasis on relative rather than absolute differences (*Lines 216-223*).
>
> Finally, the question you raise about why particular differences emerge in specific years is a good one. We note that our measure identifies a change in the volatility of hawk/dove sentiment after 2000 that aligns with the historical record and past empirical work on FOMC communication strategy (*Line 245*). Indeed, there are many other interesting specific years of interest we would like to highlight, but we kept it to this one example for the sake of brevity.

---

### Official Review · Reviewer_TSKC · 2023-08-03

**Soundness:** 4

**Excitement:**

4: Strong: This paper deepens the understanding of some phenomenon or lowers the barriers to an existing research direction.

**Missing References:**


Romer, C. D., Romer, D. H., 2004. A new measure of monetary shocks: Derivation and implications. American Economic Review 94 (4), 1055–1084.  (One of many papers in the area -- hawkish and dovish depend on the data the fed is looking at)

The cite to Blider et. al. is a good cite.  But note that is a survey article.  People have discussed the transparency of monetary policy for a long time.

**Paper Topic And Main Contributions:**

US Monetary Policy is conducted through the Federal Reserve System, specifically the "Federal Open Markets Committee" (FOMC).  The FOMC meets 8 times per year and produces a "statement" and a "transcript."  The FOMC also has an action in setting the Fed Funds Rate (not used here).   The paper uses GPT-4 model to chart the "hawk/dove" sentiment in US monetary policy "statement" (i.e., press release) and the meeting's "transcript."  The main question -- does the issued statement reflect the sentiment of the transcript of the meetings.

**Questions For The Authors:**

A: Figure 3 has NBER recessions as grey bars. It would be helpful to note that.  Related, one of the colors of the lines is also grey and that can lead to confusion.

B: Figure 3 and in general, define "hawk" and "dove."  (And a few sentences explaining why people might have a different opinion on how aggressive monetary policy should be towards inflation might help the audience.  Perhaps how this is different/similar to "sentiment".

C:  It might be helpful to include some measure of inflation over this history.  (It seems odd in 2023, but in some of those meetings, they are arguing about how to _increase_ inflation to the 2% goal).

D: Similarly, much of Fed policy in 2008 was not focussed on (just) inflation.  That was the period of "unconventional monetary policy" related to the financial crisis (2007 to 2009).

E:  What is the rationale for: "Our analysis of transcripts is of an unprecedented scale relative to past social science work."  There such a large amount of economics about monetary policy and many scholars have quoted from the transcripts.  Or, research that uses the text in the Beige Book (or other Fed forecasts books).

**Reasons To Accept:**

- The annotated dataset has lots of interesting applications
- Monetary policy is an important topic.  In particular, the transcript that documents decision-making and the statement of the "consensus" is a rich setting for NLP models.
- documenting that the transcript and statement have similar sentiment (in Fig 3b) is interesting.
- The use of the GPT is clearly explained and the fact it works well (relative to labeled data) at capturing the sentiment is interesting.

**Reasons To Reject:**

- The paper is an application of GPT-4 and not producing a new nlp model.
- The overall conclusions of the paper are not compelling. For example:  What do we learn from Figure 3b?  Why is this relevant? Does the transcript refect differences of opinion?

**Reproducibility:**

4: Could mostly reproduce the results, but there may be some variation because of sample variance or minor variations in their interpretation of the protocol or method.

**Reviewer Confidence:**

4: Quite sure. I tried to check the important points carefully. It's unlikely, though conceivable, that I missed something that should affect my ratings.

---

> ### Author Rebuttal · Authors · 2023-08-28
>
> We thank Reviewer 2 for their detailed review.  We agree that monetary policy is extremely important and that analyzing transcripts with GPT is an important application.
>
> We acknowledge that “The paper is an application of GPT-4 and not producing a new nlp model.”  While we could create a new model, it is not obvious to us that this would be preferable to creating prompts for an existing model.   We are not able to train a model as large as GPT (or any other comparable large language model).  Furthermore, use of GPT-4 makes replications easier. That said, we are organizing data for NLP purposes, which could ultimately be used in training the next generation of larger language models.  The technical language of the Federal Reserve is not likely to be found in standard common crawl resources.  We release all our annotations (both manual and by GPT), as well as the prompts.
>
> We will work to clarify in future drafts why the overall conclusions should be seen as compelling. We introduce the hypothesis that there is dissent in transcripts that is not present in statements—a key point missed by the prior, near exclusive, focus on statements.  We support this hypothesis with quotes found through manual reading of the transcripts (*Lines 132-139, 144-176*).  Figure 3b empirically shows that both the peak of hawkish and dovish sentiment occurs in the transcripts rather than the statements (*Lines 264-267*). Additionally, this entire experiment is a successful use of large language models at scale for an important social science question which itself provides additional evidence for the growing body of work emphasizing the feasibility of these approaches.
>
> To answer the questions:
> 1. We note this confusion and thank you for your feedback.
> 2. This is a good suggestion to improve readability.  (Please see *Lines 178-182* for a reference to our definition of Hawk and Dove labels). With the additional 1 page of space, we will write out the  Hansen and Kazinnik (2023) definitions in the intro and reference them in Figure 3.
> 3. We agree that it might be interesting to note how inflation changed over time, but our analysis focused on how dissent was captured, more generally, in meeting transcripts and statements. Inflation was a key topic during many of these meetings (see Figure 4 in the appendix), but changes in the inflation rate were not the main focus of our paper.  We will investigate to see if the actual inflation for the year is interesting relative to the discussions.
> 4. We think this would be excellent follow-up work.  We used inflation as an example for how dissent in the transcripts was diluted or omitted in the statement but did not exclusively evaluate Hawkishness or Dovishness in relation to inflation. (Real growth was also relevant.) We considered different areas of investigation but thought inflation would be the best starting point given the existing literature and our desire to demonstrate the scalability of large language models for this task; more niche discussions such as "unconventional monetary policy” would only apply to recent years rather than to the entire transcript history.
> 5. You inquired about our use of unprecedented scale.  We mean for this problem, no other work that has looked at every word in every transcript. One of our contributions is creating a combined data set of all three FOMC documents (statements, minutes, and transcripts) in coordination with Federal Reserve staff. With this, we were able to conduct the first analysis of multiple types of FOMC documents (statements and transcripts) from 1994 to 2016. No analysis has previously analyzed these documents in tandem or with this quantity of data. Given the venue, we hope this will encourage other work in social science.  We will clarify this in the final version.

---

### Official Review · Reviewer_dn37 · 2023-08-04

**Soundness:** 2

**Excitement:**

3: Ambivalent: It has merits (e.g., it reports state-of-the-art results, the idea is nice), but there are key weaknesses (e.g., it describes incremental work), and it can significantly benefit from another round of revision. However, I won't object to accepting it if my co-reviewers champion it.

**Paper Topic And Main Contributions:**

This paper presents a novel study to understand patterns of dissent among members on the topic inflation within the Federal Open Market Committee (FOMC) by using GPT-4.

**Questions For The Authors:**

* Could you explain what measures or systems we could use to verify the quality of the use of GPT in such an analysis (in terms of reproducibility)? I was wondering if you have also tried an open-source system that provides more transparency etc.?
* The phrase 'escalates any borderline cases' in your annotation process is a bit vague; could you provide more clarification on this aspect and the annotation process?
* Could you expand on the comparison between manual and GPT-based annotations in your work? What kinds of differences/disagreements did you mainly observe in their evaluation?

**Reasons To Accept:**

* relevance of the problem: The paper addresses a relevant research topic - understanding the dynamics of dissent within the Federal Open Market Committee, through the use of advanced technology in the form of GPT-4
* great resource: expansion of a previous work by Hansen and Kazinnik (2023), by extending the analysis to 3728 sentences (1994-2016), providing a more solid grounding for a comprehensive understanding of dissent within the FOMC

**Reasons To Reject:**

* lack of quality measure: for me, a significant drawback of the paper is the absence of quality measures for sanity checks. The use of GPT-4 is indeed promising; however, without a proper system for verifying the quality and accuracy of the results, there is a risk of drawing inaccurate conclusions. Such a measure would have ensured the reliability of the findings and added more credibility to the analysis.
* incremental nature of the paper: the paper's research is an extension of the work by Hansen and Kazinnik (2023), with the primary difference being the shift from GPT-3 to GPT-4. The contribution could be seen as relatively incremental.
* (format and depth of analysis: given its complexities and the volume of data considered, the study would benefit from a longer paper format. A more comprehensive format would allow for a more in-depth analysis, discussion of potential limitations, and more detailed contextualization of GPT-4 compared to other models) - for the short paper version, this is just an optional aspect or something for the appendix
* annotation: I think the study could be enriched further by an extensive comparison between manual and GPT-based annotations, examining aspects such as error types and agreement or others to quantify any potential discrepancies. Furthermore, a more detailed description of the annotation process would be beneficial, as terms like 'escalates any borderline cases' are somewhat ambiguous and leave room for clarification.

**Reproducibility:**

2: Would be hard pressed to reproduce the results. The contribution depends on data that are simply not available outside the author's institution or consortium; not enough details are provided.

**Reviewer Confidence:**

3: Pretty sure, but there's a chance I missed something. Although I have a good feel for this area in general, I did not carefully check the paper's details, e.g., the math, experimental design, or novelty.

---

> ### Author Rebuttal · Authors · 2023-08-28
>
> We thank Reviewer 1 for the clarifying questions and constructive criticism.  We agree that studying dissent on inflation in the Federal Reserve is highly relevant and that creating a resource for the NLP community would allow us to better understand it.
>
> We wholeheartedly agree with the need for quality annotation (*Reasons to Reject #1, #4, Question #2*).  We manually review every FOMC issued statement as a whole, evaluating the overall Hawkishness or Dovishness of the document (*Section 2.3*). We then compare these manual annotations to the GPT-based annotation. Of the two GPT methods, prompting to evaluate statements based on the average of its individual sentences and prompting it to evaluate the statement as a whole, the latter matched more closely to the manual annotation. The main disagreement we observed was that the GPT evaluation would label sentences as more neutral than the manual analysis concluded.   Additionally, we manually read a transcript (*Lines 185-188, Appendix C, Figure 4*).  Since the transcript is hundreds of pages in length, this required five hours and specialized knowledge of the Federal Reserve, making the process intractable to repeat for all transcripts in the same way as for statements.
>
> Escalation on borderline cases was intentionally vague in order to preserve anonymity, and was done from a sociology undergraduate to a graduate student. Results of the annotation were discussed with a leading expert, omitted in this version for the sake of anonymity.  Most cases are obvious so this allows scalability while ensuring a genuine gold standard (15 cases needed escalation, of which 2 were escalated to the leading expert).  We strongly agree with the author’s request to include a more detailed analysis of disagreements between GPT and human annotation. We will take this advice and prioritize explaining the annotation process and the analysis in the additional 1-page.
>
> We ground our work in existing literature, including Hansen and Kzinnik (2023).  Given the close proximity of their contributions and ours, we agree with the reviewer that it is important to emphasize our distinct contribution which we will do so in the revised version.  In short, Hansen and Kzinnik test the application of multiple models, including GPT-3 and GPT-4, on several different language tasks using FOMC materials. We do a robust analysis on a more specific task - quantifying dissent among the FOMC. Where they annotate only 500 individual statements selected at random from statements over a 10 year period and have no time dimension in their analysis, we go much further. First, we demonstrate an improvement over their methodology (Figure 3A) by evaluating the statement as a whole. Second, we do this at a considerably larger scale, annotating all 3728 individual sentences in statements from 1994-2016 compared to their 500 sentences at random.  Third, we create a gold standard to assess their methodology using non-trivial manual annotation. Fourth and most important, we introduce a previously unstudied dimension and dataset, by analyzing transcripts and minutes, which are magnitudes larger than just the statements and evaluating changes over time.  While the statements are short enough that economists could plausibly use manual annotation to investigate, we demonstrate the feasibility of using GPT-4 to vastly expand the quantitative analysis of Federal Reserve deliberations. As a side benefit, our methodology demonstrates a powerful new scale of analysis for social science research more broadly.
>
> We agree that a longer paper format would be excellent (*Reasons to Reject #3*).  However, since this is building on past work, we focused our contribution on dissent within the Federal Reserve, which should be possible to convey with the additional page for the final version. To deal with the page limit, we optimized space in this submission and submitted a 5 page appendix (with real examples of our data, technical details of prompting, and a sanity check to support the findings of our manual analysis of the transcripts).  We will include any additional details that arise from the additional requested analysis.
>
> We understand and share your interest in open-source alternatives (*Question #1*). A human expert on our team examined GPT-4 generated labels, and found that in a sample of 25, our expert agreed with 19 labels with high confidence, and 22 labels with at least moderate confidence. This, as well as recent work that successfully uses GPT-4 for classification (References [1][2][3] below) gave us confidence in its quality. Reproducibility is still an issue; although we set Temperature and Top-P to 0, we understand that randomness in GPU computations still causes variability in responses. Additionally, OpenAI models change over time. As such, it may not be possible to exactly reproduce these results. We experimented with open source models such as FLAN-T5-XXL, but found that the limited performance and context length rendered it insufficient for our task. We are optimistic about newer open models such as Llama 2, but at the time of submission we feel that GPT-4 is the strongest viable option.
>
> References:
>
> [1] OpenAI. GPT-4 Technical Report. 2023.
>
> [2] Liu et al. Summary of ChatGPT-Related Research and Perspective Towards the Future of Large Language Models. 2023.
>
> [3] Guan et al. CohortGPT: An Enhanced GPT for Participant Recruitment in Clinical Study. 2023.

---

### Meta-Review · Area_Chair_SAz2 · 2023-09-19

**Recommendation:** 3

**Metareview:**

The paper presents a study leveraging GPT-4 to analyze the patterns of dissent within the Federal Open Market Committee (FOMC) on the subject of inflation. The paper addresses an important research problem, i.e., understanding the dynamics within FOMC, and employs a novel approach using a state-of-the-art model, GPT-4, which is noteworthy. The paper presents an annotated dataset that can be valuable for a variety of applications, including economics, finance, and NLP research. The most glaring weakness is the lack of quality measures or sanity checks to ensure the reliability of the analysis. Given the high-stakes nature of the topic, this is crucial for both academic and policy implications. The research primarily differs from the previous work by the transition from GPT-3 to GPT-4. This could be seen as an incremental step, diminishing the paper's overall impact. In light of the identified merits and shortcomings, the collective judgment leans toward accepting the paper for the "Findings" section.

---

### Decision · Program_Chairs · 2023-10-07

**Decision:**

Accept-Findings

**Comment:**

The paper presents a study leveraging GPT-4 to analyze the patterns of dissent within the Federal Open Market Committee (FOMC) on the subject of inflation. The paper addresses an important research problem, i.e., understanding the dynamics within FOMC, and employs a novel approach using a state-of-the-art model, GPT-4, which is noteworthy. The paper presents an annotated dataset that can be valuable for a variety of applications, including economics, finance, and NLP research. The most glaring weakness is the lack of quality measures or sanity checks to ensure the reliability of the analysis. Given the high-stakes nature of the topic, this is crucial for both academic and policy implications. The research primarily differs from the previous work by the transition from GPT-3 to GPT-4. This could be seen as an incremental step, diminishing the paper's overall impact. In light of the identified merits and shortcomings, the collective judgment leans toward accepting the paper for the "Findings" section.